# The Framework of Cross-Domain and Model Adversarial Attack against Deepfake

Haoxuan Qiu, Yanhui Du and Tianliang Lu *

College of Information and Cyber Security, People's Public Security University of China, Beijing 100038, China; qiuhaoxuan78@gmail.com (H.Q.); duyanhui@ppsuc.edu.cn (Y.D.)
* Correspondence: lutianliang@ppsuc.edu.cn

**Abstract:** To protect images from the tampering of deepfake, adversarial examples can be made to replace the original images by distorting the output of the deepfake model and disrupting its work. Current studies lack generalizability in that they simply focus on the adversarial examples generated by a model in a domain. To improve the generalization of adversarial examples and produce better attack effects on each domain of multiple deepfake models, this paper proposes a framework of Cross-Domain and Model Adversarial Attack (CDMAA). Firstly, CDMAA uniformly weights the loss function of each domain and calculates the cross-domain gradient. Then, inspired by the multiple gradient descent algorithm (MGDA), CDMAA integrates the cross-domain gradients of each model to obtain the cross-domain perturbation vector, which is used to optimize the adversarial example. Finally, we propose a penalty-based gradient regularization method to pre-process the cross-domain gradients to improve the success rate of attacks. CDMAA experiments on four mainstream deepfake models showed that the adversarial examples generated from CDMAA have the generalizability of attacking multiple models and multiple domains simultaneously. Ablation experiments were conducted to compare the CDMAA components with the methods used in existing studies and verify the superiority of CDMAA.

**Keywords:** deepfake; adversarial attack; generalization; CDMAA



## 1. Introduction

Deepfake [1] constructs generator models based on generative adversarial networks (GANs) to forge images. Receiving real images as input, the deepfake model can output fake images by, for example, changing hair color. Deepfake has played an important role in the entertainment and culture industry, bringing many conveniences to life and work. However, malicious users may take advantage of this technology to produce fake videos and news, misleading face recognition systems and seriously disrupting the social order [2,3].

In order to cope with deepfake tampering with images, a large number of studies focus on constructing deepfake detection models [4–10], which can detect whether an image is faked. However, this technology can only ensure the authenticity of the image, but it cannot guarantee the integrity of the image. Moreover, even if an image is confirmed a fake, negative impacts are caused on the people concerned or on society because the image has already been widely circulated. More direct interventions should therefore be taken to ensure that images are not tampered with though deepfake from the source.

Some studies propose the use of the adversarial attack [11] to interfere with the work of the deepfake model. The main idea of the adversarial attack is adding a perturbation, imperceptible to the naked eye, to the original example, generating the adversarial example, which can mislead the deep learning models to produce a quite different output. The adversarial attack was originally used to destroy security systems such as face recognition, which posed a huge challenge to the security of deep learning models. However, if the

object of an adversarial attack is turned into a malicious model such as through deepfake, the meaning of the adversarial attack becomes dramatically opposite: to disrupt the normal operation of malicious models to guarantee information security. As shown in Figure 1, the specific operation involves adding a perturbation, imperceptible to the naked eye, to an image when users post the image online so that when the attacker obtains the image, the fake version generated through deepfake will have obvious distortions or deformations, which can be easily identified as forgery.

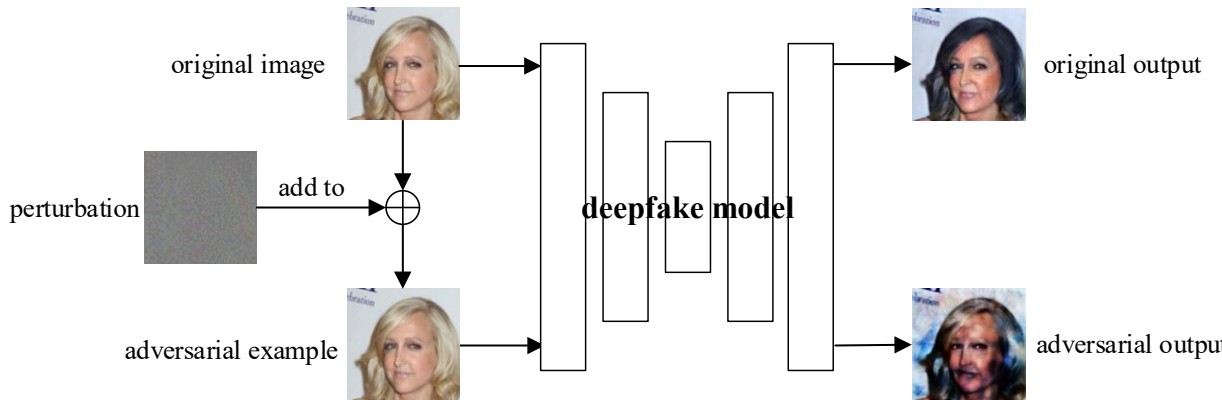

**Figure 1.** Example of adversarial attack on deepfake. The output image with the input of an adversarial example is significantly distorted, while the perturbation added to the original image is visually undetectable.

In the current study, however, the generalization of the adversarial attack against the deepfake model is very limited: an adversarial example generated for a specific deepfake model is unable to produce equal attack effect on other models [12]; furthermore, even in the same model, an adversarial example generated in a particular domain cannot achieve effective attack in other domains [13] (by setting the corresponding conditional variables, deepfake models can generate multiple domains of forged images, such as the hair color or the gender of the face image). Without the knowledge of what deepfake model will be employed or what conditional variables will be set to tamper with images, the adversarial attack methods currently studied have great limitations in practice.

In order to improve the generalization of the adversarial attack, that is, to generate the adversarial samples in each domain of multiple models of given images, this paper proposes a framework of Cross-Domain and Model Adversarial Attack (CDMAA): Any gradient-based adversarial example generation algorithm can be used for an adversarial attack, such as the I-FGSM [14]. In the backpropagation phase, the algorithm uniformly weights the loss function with different condition variables in the model to extend the generalization of the adversarial example between various domains. The Multiple Gradient Descent Algorithm (MGDA) [15] is used to calculate the weighted sum of the gradients of each model to ensure the generalization of adversarial examples between various models. Finally, we propose a penalty-based gradient regularization method to further improve the success rate of adversarial attacks. CDMAA can expand the attack range of the generated adversarial example and ensure that the images are not tampered with and forged by multiple deepfake models.

## 2. Related Work

According to the category of model input, some deepfake models input random noise to synthesize images which were entirely non-existent before [16], such as ProGAN [17], StyleGAN [18], etc. Other deepfake models input real images to achieve the image translation from domain to domain. For example, StarGAN [19], AttGAN [20] and STGAN [21] can translate the facial images in domains by setting different conditional variables, such as hair color, age, etc. Unsupervised models, such as CycleGAN [22] and U-GAT-IT [23],

can only translate images to a single domain, which can be considered a special case of multi-domain translation models with a total domain number of 1. This paper focuses on image translation deepfake models and performs the adversarial attack on them to interfere with their normal functions and protect real images from being tampered with.

The adversarial attack was initially applied to classification models [24]. Goodfellow et al. proposed the Fast Gradient Sign Method (FGSM) [25]. The FGSM sets the distance between the model output of the adversarial example and the model output of the original example as the loss function. The gradient of the loss function with respect to the input indicates a direction where the output difference between the adversarial example and the original example ascends fastest. Therefore, the FGSM adds the gradient in the original example to generate an effective adversarial example. Kurakin et al. proposed the iterative FGSM (I-FGSM) [14], which iteratively performs gradient backpropagation to reduce the step size of updating adversarial examples and improving their efficiency. Many studies have since proposed various adversarial attack algorithms to optimize the efficiency of adversarial attacks, such as PGD [26], which uses random noise to initialize the adversarial examples, the MI-FGSM [27], which uses momentum to update the gradient, and APGD [28], which automatically decreases the step size.

Kos et al. [29] first extended adversarial attacks to generation models. Yeh et al. [30] first proposed to attack the deepfake model. They used PGD to generate adversarial examples against CycleGAN, pix2pix [31], etc., which can distort the output of these models. Lv et al. [32] proposed that higher weight should be given to the face part of the images when calculating the loss function so that the output distortion generated by the adversarial examples is concentrated on the face to achieve a better effect of interfering deepfake models. Dong et al. [33] explored the adversarial attacks on encoder–decoder-based deepfake models and proposed to use the loss function with respect to latent variables in encoders to generate the adversarial examples. These studies generate adversarial examples only for certain models and do not take into account that models can output fake images of different domains by setting different condition variables, so the generalization of adversarial attacks is quite limited.

Ruiz et al. [13] considered the generalizability of adversarial attacks across different domains. They verified that the adversarial example generated in a particular domain cannot achieve effective attack in other domains of the model and proposed two methods of iterative traversal and joint summation to generate adversarial examples that are effective for each domain. However, they did not consider the generalization between different models of the adversarial examples. Since the differences between models are much larger than the differences between domains within models, the simple method of iterative traversal or joint summation cannot be equally effective for attacks between different models.

Fang et al. [34] considered the generalizability of adversarial attacks across models. They verified that the adversarial examples against a particular model are ineffective in attacking other models and proposed a method of weighting the loss functions of multiple models to generate adversarial examples against multiple deepfake models, where the weighting factors are found by a line search. However, the tuning experiments are extremely tedious because the weighting coefficients need to be found in $J - 1$ dimensional parameter space, where $J$ denotes the number of models. In addition, the coefficients need to be adjusted when attack models are changed, which is quite inefficient.

Compared with the existing work, this paper focuses on extending the generalization of adversarial examples across various domains and models and proposes a framework of CDMAA. CDMAA can generate adversarial examples that can attack multiple deepfake models under all condition variables with higher efficiency.

## 3. CDMAA

In this paper, we use the I-FGSM as the basic adversarial attack algorithm to introduce the CDMAA framework. In the model forward propagation phase, we generate the cross-domain loss function of each model by uniformly weighting the loss function corresponding



to each conditional variable. In the phase of model backward propagation to calculate the gradient, we use the MGDA to generate a cross-model perturbation vector from the gradient of each cross-domain loss function. The perturbation vector is used to iteratively update the adversarial example to improve its generalizability across multiple models and domains.

### 3.1. I-FGSM Adversarial Attack Deepfake Model

Given an original image $x$, its output of the deepfake model is $G(x, c)$, where $G$ denotes the deepfake model and $c$ denotes the conditional variable. The I-FGSM generates the adversarial example $\widetilde{x}$ by the following steps:

$$x_0 = x \tag{1}$$

$$x_{t+1} = clip(x_t + a \cdot sign(\nabla_{x_t} L(G(x_t, c), G(x, c)))) \tag{2}$$

where $x_t$ denotes the adversarial example after $t$ iterations, $t$ does not exceed the number of iteration steps $T$, $a$ denotes the step size, $sign$ is the symbolic function, $\varepsilon$ denotes the perturbation range and the $clip$ function restricts the size of the adversarial perturbation not to exceed the perturbation range in $l_p$-norm, i.e.,

$$\|x_t - x\|_\infty \le \varepsilon \tag{3}$$

so that the difference between the adversarial example and the original image is sufficiently small to ensure that the original image is not significantly modified, $L$ denotes the loss function, which uses the mean squared loss (MSE) to measure the distance between the output of the adversarial example $G(x_t, c)$ and the output of the original image $G(x, c)$ [30]:

$$L(G(x_t, c), G(x, c)) = \frac{1}{D} \|G(x_t, c), G(x, c)\|_2 \tag{4}$$

where $D$ denotes the dimensionality of the model output, i.e., the $length \cdot width \cdot channels$ of the output image.

The adversarial example is updated towards the optimization goal:

$$\widetilde{x} = \underset{\widetilde{x}}{\operatorname{argmax}} L(G(\widetilde{x}, c), G(x, c)) \tag{5}$$

The generated adversarial example is considered to have successfully attacked the deepfake model $G$ under the condition variable $c$ when the loss function $L$ keeps increasing and reaches a certain threshold $\tau$, i.e., the output image has a sufficiently noticeable distortion.

### 3.2. Cross-Domain Adversarial Attack

To extend the generalizability of the adversarial examples across various domains of the model, i.e., the optimization objective (5) is modified as

$$\widetilde{x} = \underset{\widetilde{x}}{\operatorname{argmax}} \{L(G(\widetilde{x}, c_i), G(x, c_i)) | i = 1 \dots K\} \tag{6}$$

where $c_i$ denotes the $i$th conditional variable of model $G$ and $K$ denotes the total number of conditional variables.

The gradient for each of the loss functions in the above optimization objectives is calculated by

$$grad_i = \nabla_{x_t} L(G(x_t, c_i), G(x, c_i)) \tag{7}$$

where $grad_i$ indicates the optimization direction for maximizing the distortion of the output of model $G$ with condition variable $c_i$ for the current adversarial example $x_t$.

Since the backbone network is fixed in the model, changing only the condition variables has less impact on the model output, resulting in the loss functions $L_i$ and their

corresponding gradients $grad_i$ of different condition variables being very similar, i.e., the $grad_i$ have approximately the same direction, as shown in Figure 2a. Therefore, we integrate a cross-domain gradient $grad$ by simply uniformly weighting $grad_i$:

$$grad = \frac{1}{K}\sum_{i=1}^{K} grad_i = \frac{1}{K}\sum_{i=1}^{K} \nabla_{x_t} L(G(x_t, c_i), G(x, c_i)) \tag{8}$$

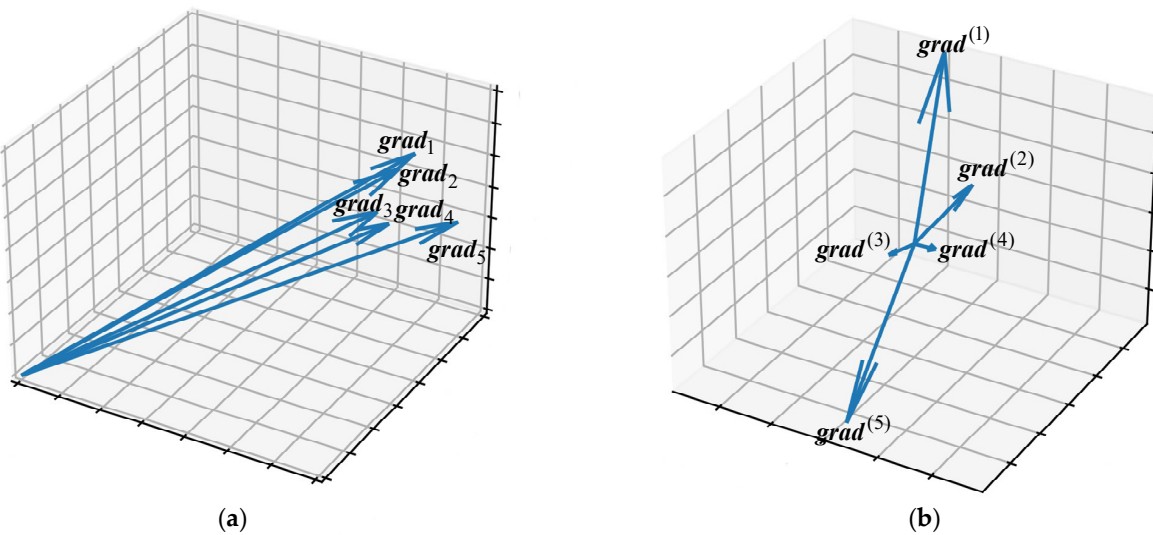

(**a**)                       (**b**)

**Figure 2.** Visualization of gradients between the domains or models (Case the number of gradients is 5). Gradients between domains (**a**) tend to have similar directions and norms since they are from the same model. Gradients between models (**b**) are largely different in their directions and norms.

$grad$ is obtained by integrating the loss functions corresponding to each conditional variable so that they indicate a common direction of maximizing the loss function of each domain. Using $grad$ to update the adversarial example can achieve the optimization objective of (6).

Consider the following equation:

$$\frac{1}{K}\sum_{i=1}^{K} \nabla_{x_t} L(G(x_t, c_i), G(x, c_i)) = \nabla_{x_t}\left(\frac{1}{K}\sum_{i=1}^{K} L(G(x_t, c_i), G(x, c_i))\right) = \nabla_{x_t}\left(\frac{1}{K}\sum_{i=1}^{K} L_i\right) \tag{9}$$

That is, we can uniformly weight the loss function $L_i$ corresponding to each condition variable $c_i$ to obtain a cross-domain loss function $\sum_{i=1}^{K} L_i$ and then calculate the gradient of it with respect to $x_t$, which is the cross-domain gradient $grad$. It ensures that only one backpropagation step is performed for each model so that time consumption is reduced.

### 3.3. Cross-Model Adversarial Attack

We further extend the generalizability of the adversarial examples across models, i.e., the optimization objective in (6) is modified:

$$\tilde{x} = \underset{\tilde{x}}{\mathrm{argmax}}\left\{L\left(G^{(j)}(\tilde{x}, c_i), G^{(j)}(x, c_i)\right)\Big| i = 1 \dots K_j, j = 1 \dots J\right\} \tag{10}$$

where $G^{(j)}$ denotes the $j$th deepfake model and $J$ denotes the total number of deepfake models. The group of cross-domain gradients has been obtained from Section 3.2:

$$grad = \left\{grad^{(1)}, \dots, grad^{(J)}\right\} \tag{11}$$

where $\textbf{\textit{grad}}^{(j)}$ denotes the cross-domain gradient of the *j*th model. Considering that these gradients come from different models, the large differences between models lead to a low similarity between the gradients, as shown in Figure 2b. Simply iterative traversing or uniform weighting these gradients is prone to a large fluctuation in the optimization process and generates an ineffective adversarial example [35].

In order to derive a cross-model perturbation vector $\textbf{\textit{w}}$ from the group of gradients $\textbf{\textit{grad}}$ to update the adversarial example, the CDMAA framework draws on the idea of the Multiple Gradient Descent Algorithm (MGDA) to give an idea for finding $\textbf{\textit{w}}$:

$$\textbf{\textit{w}} = \arg\min_{\textbf{\textit{u}} \in \overline{U}} \|\textbf{\textit{u}}\|_2^2 \tag{12}$$

The space $\overline{U}$ that the vector $\textbf{\textit{u}}$ values in satisfies:

$$\overline{U} = \left\{ \textbf{\textit{u}} = \sum_{j=1}^{J} a^{(j)} \cdot \textbf{\textit{grad}}^{(j)} \middle| a^{(j)} \geq 0, \forall j; \sum_{j=1}^{J} a^{(j)} = 1 \right\} \tag{13}$$

**Theorem 1.** *The solution $\textbf{\textit{w}}$ in (12) is the optimization direction in which the loss function corresponding to each model is increasing for the current adversarial example, i.e., it satisfies:*

$$\textbf{\textit{w}} \cdot \textbf{\textit{grad}}^{(j)} > 0, \forall j \tag{14}$$

**Proof.** Equation (12) is equivalent to the following optimization problem:

$$\textbf{\textit{a}}_0 = \underset{\textbf{\textit{a}} \in \overline{A}}{argmin} \left\| \sum_{j=1}^{J} a^{(j)} \cdot \textbf{\textit{grad}}^{(j)} \right\|_2^2, \ s.t. \overline{A} = \left\{ \textbf{\textit{a}} = \left( a^{(1)}, \dots, a^{(J)} \right) \in \mathbb{R}_+^J \middle| \sum_{j=1}^{J} a^{(j)} = 1 \right\} \tag{15}$$

To solve this extreme value problem of a multivariate function under the linear constraint, construct the Lagrange function:

$$L(\textbf{\textit{a}}, \lambda) = \|\textbf{\textit{u}}\|_2^2 + \lambda \left( \sum_{j=1}^{J} a^{(j)} - 1 \right) \tag{16}$$

since $\textbf{\textit{u}} = \sum_{j=1}^{J} a^{(j)} \cdot \textbf{\textit{grad}}^{(j)}$; hence $\frac{\partial \textbf{\textit{u}}}{\partial a^{(j)}} = \textbf{\textit{grad}}^{(j)}$ and

$$\frac{\partial L(\textbf{\textit{a}}, \lambda)}{\partial a^{(j)}} = \frac{\partial \|\textbf{\textit{u}}\|_2^2}{\partial a^{(j)}} + \lambda = 2\textbf{\textit{u}} \cdot \frac{\partial \textbf{\textit{u}}}{\partial a^{(j)}} + \lambda = 2\textbf{\textit{u}} \cdot \textbf{\textit{grad}}^{(j)} + \lambda \tag{17}$$

According to the Lagrange multiplier, the equation

$$\frac{\partial L(\textbf{\textit{a}}, \lambda)}{\partial a^{(j)}} = 0, \forall j \tag{18}$$

is a necessary condition for $\textbf{\textit{w}}$ obtaining the minimum of $\textbf{\textit{u}}$; hence,

$$\textbf{\textit{w}} \cdot \textbf{\textit{grad}}^{(j)} = -\frac{\lambda}{2}, \forall j \tag{19}$$

considering that

$$\|\textbf{\textit{w}}\|_2^2 = \textbf{\textit{w}} \cdot \textbf{\textit{w}} = \textbf{\textit{w}} \cdot \sum_{j=1}^{J} \left( a^{(j)} \cdot \textbf{\textit{grad}}^{(j)} \right) = \sum_{j=1}^{J} a^{(j)} \textbf{\textit{w}} \cdot \textbf{\textit{grad}}^{(j)} \tag{20}$$

In the actual adversarial attack scenario, since the dimension $D$ is much larger than the number $J$ of the gradients in $grad$, it is almost impossible for these gradients to be linearly dependent. Hence, their linear combination $w \neq 0$ and

$$\|w\|_2^2 > 0 \tag{21}$$

Uniting $\sum\limits_{j=1}^{J} a^{(j)} = 1, w \cdot grad^{(j)} = -\frac{\lambda}{2}$, there is

$$-\frac{\lambda}{2} > 0 \tag{22}$$

Simultaneously, (19), (20) and (14) are proven. □

Since the vector product of $w$ and the gradient of all model loss functions is positive, optimizing the adversarial example with $w$ ensures the whole improvement of loss functions in each model, i.e., the optimization objective (10), which can improve the generalization of the adversarial examples in various models.

*3.4. Gradient Regularization*

In Section 3.3, if the gradients group $grad$ is regularized as

$$grad_{nor}^{(j)} = grad^{(j)} / S^{(j)} \tag{23}$$

and then the MGDA is used on the regularized gradients group $grad_{nor}$ to find a perturbation vector $w$, the result of (14) holds because

$$w \cdot grad^{(j)} = w \cdot grad_{nor}^{(j)} \cdot S^{(j)} > 0 \tag{24}$$

where $S^{(j)} > 0$ is the regularization factor.

Common regularization methods include *L*2 regularization: $S^{(j)} = \|grad^{(j)}\|_2$, which scales the gradients to the unit vector; logarithmic gradients regularization: $S^{(j)} = L^{(j)}$, which reduces the gradients by the factor of their corresponding loss function value.

Due to the large difference in norms of each of the cross-domain gradients $grad^{(j)}$ which are calculated from various models, the resulting vector $w$ is expected to be mostly influenced by the gradients of small norms in $grad^{(j)}$. In addition, without some constraints and guidance methods, the generated adversarial example will form an obvious "attack preference" due to the different vulnerability of deepfake models, only achieving high attack effect on the vulnerable models, which eventually leads to a large difference of different models.

To lead the cross-model perturbation vector in the direction of improving the effectiveness of attacks on models that are not vulnerable to adversarial attacks and maximize the success rate of adversarial attacks against all models, we propose a penalty-based gradient regularization method:

$$S^{(j)} = \frac{1}{\max\left(L^{(j)}, \varsigma\right)} \tag{25}$$

where $L^{(j)}$ denotes the cross-domain loss function of the $j$th model and $\varsigma$ is a very small positive number to prevent the zero-denominator error of $S^{(j)}$ when $L^{(j)} = 0$. (The value of the loss function $L^{(j)}$ is 0 inevitably in the first iteration of the I-FGSM since the current adversarial example is the same as the original image).

The significance of using this gradient regularization is as follows:

According to (19), the $w$ derived from the regularized gradient $grad_{nor}$ satisfies

$$w \cdot \frac{grad^{(j)}}{S^{(j)}} = w \cdot grad_{nor}^{(j)} = -\frac{\lambda}{2}, \forall j \tag{26}$$

Consider the first-order Taylor expansion of the loss function $L^{(j)}(x_t) = L\left(G^{(j)}(x_t, c), G^{(j)}(x, c)\right)$ at the $t$th iteration:

$$\Delta L^{(j)} = L^{(j)}(x_t + a \cdot sign(w)) - L^{(j)}(x_t) \approx L^{(j)}(x_t + a \cdot w) - L^{(j)}(x_t)$$
$$\approx \nabla_{x_t} L^{(j)}(x_t) \cdot (a \cdot w) = a \cdot w \cdot grad^{(j)} \tag{27}$$

where the first approximately equal sign ignores the effect of taking the sign function for $w$ on the result, and the second approximately equal sign ignores the remainder of the first-order Taylor formula for approximation.

Uniting (26) and (27), there is

$$\Delta L^{(j)} \approx a \cdot w \cdot grad^{(j)} = -\frac{a\lambda}{2} \cdot S^{(j)} = -\frac{a\lambda}{2} \cdot \frac{1}{L^{(j)}} \tag{28}$$

The last equal sign of the above equation can be held by taking a sufficiently small $\varsigma$. Since $-\frac{a\lambda}{2}$ is a constant, each model's value of the loss function change $\Delta L^{(j)}$ is inversely proportional to their current corresponding loss function value $L^{(j)}$, implying that the smaller the value of the loss function, the larger the optimization gain can be obtained. In practical adversarial attacks, the adversarial examples achieve successful attacks on some vulnerable models after a small number of iterative steps, as their corresponding loss functions have reached the threshold. It is meaningless to further improve these loss functions. Using this regularization can make the adversarial example mainly optimized in the direction of improving the loss functions that have not reached the threshold, which can improve the attack effect on their corresponding models and pursue a higher comprehensive attack success rate.

### 3.5. CDMAA Framework

In summary, this paper proposes a framework of adversarial attacks on multiple domains of multiple models simultaneously. Using the I-FGSM adversarial attack algorithm as an example, the procedure of CDMAA is as follows (Algorithm 1):

---

**Algorithm 1** CDMAA

---

Input: original image $x$, iterative steps $T$, perturbation magnitude $\varepsilon$, step size $a$, deepfake model group $G^{(j)}, j = 1, \ldots, J$
Output: adversarial example $\widetilde{x}$
Initialization: $x_0 = x$

1.　**For** $t = 0$ **to** $T - 1$ **do**
2.　**For** $j = 1$ **to** $J$ **do**
3.　$L^{(j)} = \frac{1}{K^{(j)}} \sum\limits_{i=1}^{K^{(j)}} L\left(G^{(j)}(x_t, c_i), G^{(j)}(x, c_i)\right)$
4.　$grad^{(j)} = \nabla_{x_t} L^{(j)}$
5.　$grad_{nor}^{(j)} = grad^{(j)} / \frac{1}{\max\left(L^{(j)}, \varsigma\right)}$
6.　**End for**
7.　$w = \underset{a \in \overline{A}}{argmin} \left\|\left| \sum\limits_{j=1}^{J} a^{(j)} \cdot grad_{nor}^{(j)} \right\|\right\|_2^2$
8.　$x_{t+1} = clip(x_t + a \cdot sign(w))$
9.　**End for**
10.　$\widetilde{x} = x_T$

---

Step 3 is the use of the uniformly weighting method to obtain the cross-domain loss function, which is sufficiently effective due to the similarity of the gradients between domains (Section 3.2). It needs only one backpropagation step to calculate gradient in

the following step, while the MGDA needs $K^{(j)}$ backpropagation steps to calculate the gradients in each domain, thus ensuring good efficiency.

Step 7 is the use of the MGDA to obtain the cross-model perturbation vector, where a simple uniformly weighting method is less effective due to the low similarity of gradients between models (Section 3.3). Therefore, the MGDA is used to achieve better attacks at the expense of time. We use the Frank–Wolfe method [36] to approximately calculate the minimal norm vector in the convex hull of $grad_{nor}$, which has a well convergence in such cases that the number of dimensions is much larger than the number of vectors [35,37].

Figure 3 shows the overview of CDMAA. It is noted that CDMAA is not necessarily applied on the I-FGSM, although this paper uses the I-FGSM to introduce CDMAA. The main idea of CDMAA is to obtain a perturbation vector from gradients in multiple domains and models and then update the adversarial examples to ensure their ability of attacking multiple models and domains. Therefore, CDMAA can be applied to any gradient-based adversarial attack algorithms, such as the MI-FGSM and APGD.

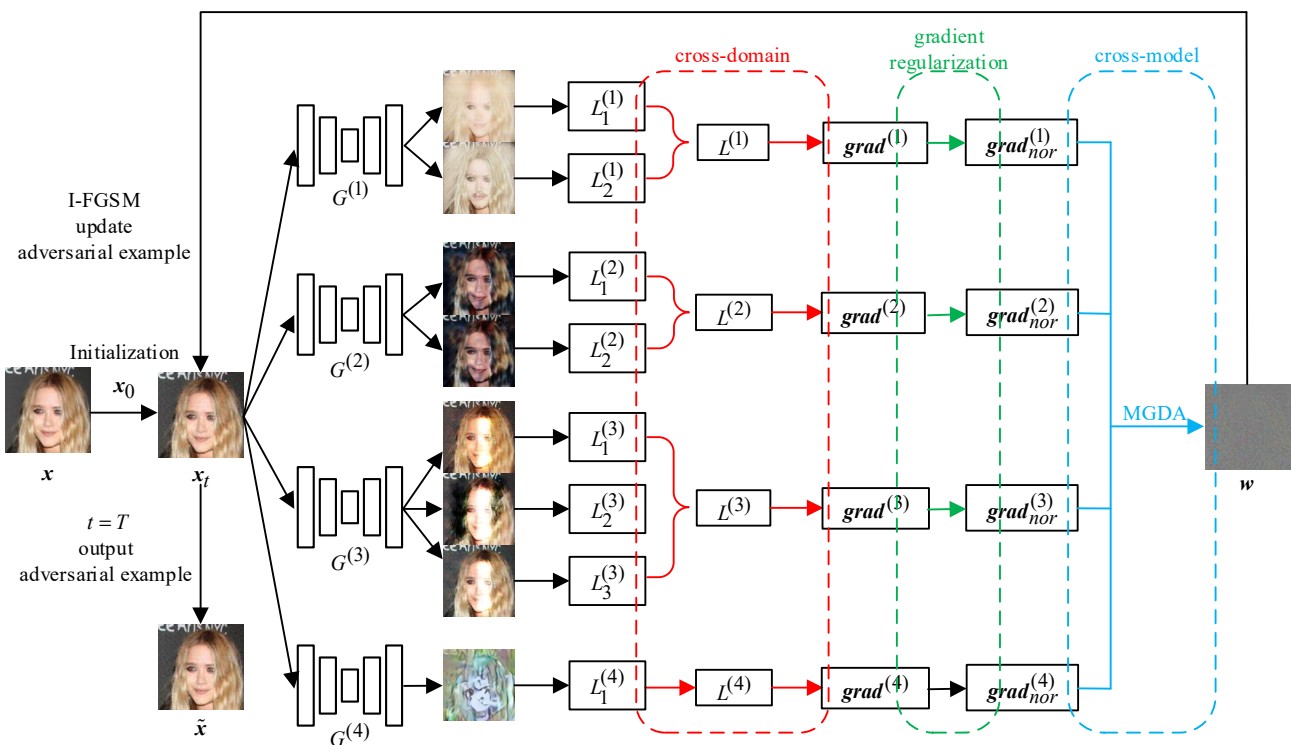

**Figure 3.** Overview of CDMAA.

## 4. Experiment and Analysis

To verify the effectiveness of the proposed CDMAA framework, we conduct adversarial attack experiments against deepfake models and analyze the results. In Section 4.1, we introduce the deepfake model, hyper-parameters and evaluation criteria used in the experiments. In Section 4.2, we use CDMAA to attack four models at the same time and show the result of adversarial attacks. In Section 4.3, we conduct ablation experiments to show the impact of CDMAA components on the attack and compare with the methods used in the existing work.

### 4.1. Deepfake Models, Hyper-Parameters and Evaluation Metrics

We prepared four deepfake models—StarGAN, AttGAN, STGAN and U-GAT-IT—for the adversarial attack experiments, which are chosen in similar existing work [12,13,34]. StarGAN and AttGAN adopt the officially provided pre-training models, which are training respectively in five domains—black hair, blonde hair, brown hair, gender and age—as well as in 13 domains, such as bald head and bangs on the celebA dataset. STGAN uses

the model trained on the celebA dataset in five domains—bangs, glasses, beard, slightly opened mouth and pale skin—which are rare attributes in original images. We selected these domains to prevent the possibility that the STGAN output will be the same as the input when the original picture already contains the attributes of the corresponding domains; in which case, the experiment results will be affected since the model is unable to effectively forge the images even without adversarial examples [34]. U-GAT-IT realizes the translation of images from a single domain to another, so it can be regarded as a special case of multi-domain deepfake when the total number of conditional variables $K_{\text{U-GAT-IT}} = 1$. To unify the dataset in the experiments, we used the U-GAT-IT model to translate from celebA to the anime dataset, which is trained on official codes.

The adversarial attack algorithm uses the I-FGSM, in which the hyper-parameters refer to the settings in the existing work, $T = 80, \varepsilon = 0.05$ and $a = 0.01$, except where noted. The test data are $N = 100$ randomly sampled pictures in the celebA dataset (ensure that the pictures used in each contrast experiment are the same).

The value of the loss function

$$L_i^{(j)}\left(\widetilde{\pmb{x}}^{(n)}, \pmb{x}^{(n)}\right) = L\left(G^{(j)}\left(\widetilde{\pmb{x}}^{(n)}, c_i\right), G^{(j)}\left(\pmb{x}^{(n)}, c_i\right)\right) \tag{29}$$

is used to quantify the output distortion of the $n$th adversarial example $\pmb{x}^{(n)}$ to model $G^{(j)}$ under the condition variable $c_i$. The following evaluation criteria [13] are considered to evaluate the effectiveness of the adversarial attack:

$$avg\_L^{(j)} = \frac{1}{N \cdot K_j} \sum_{n=1}^{N} \sum_{i=1}^{K_j} L_i^{(j)}\left(\widetilde{\pmb{x}}^{(n)}, \pmb{x}^{(n)}\right) \tag{30}$$

$$attack\_rate^{(j)} = \frac{1}{N \cdot K_j} \sum_{n=1}^{N} \sum_{i=1}^{K_j} 1_{L_i^{(j)}\left(\widetilde{\pmb{x}}^{(n)}, \pmb{x}^{(n)}\right) > \tau} \tag{31}$$

where $avg\_L^{(j)}$ represents the average value of the loss function of $N$ adversarial examples in each domain for model $G^{(j)}$ and $attack\_rate^{(j)}$ represents the proportion of the loss function of $N$ adversarial examples in each domain reaching the threshold $\tau = 0.05$ [13] for Model $G^{(j)}$, i.e., the attack success rate.

### 4.2. CDMAA Adversarial Attack Experiment

We used CDMAA framework to attack the four deepfake models StarGAN, AttGAN, STGAN and U-GAT-IT at the same time. The results are shown in Table 1:

**Table 1.** Adversarial attack deepfake models with CDMAA. The results of five groups of experiments are listed as five columns. (a) Basic group. Set the default attack algorithm (I-FGSM) and parameters ($T = 80, \varepsilon = 0.05, a = 0.01$). (b) Increase $T$ to 100 times. (c) Expand $\varepsilon$ to 0.06. (d) Use the MI-FGSM as the attack algorithm. (e) Use APGD as the attack algorithm.

| | (a) I-FGSM | | (b) I-FGSM $T$=100 | | (c) I-FGSM $\varepsilon$=0.06 | | (d) MI-FGSM | | (e) APGD | |
| | Avg_L ↑ | Attack_Rate ↑ | Avg_L ↑ | Attack_Rate ↑ | Avg_L ↑ | Attack_Rate ↑ | Avg_L ↑ | Attack_Rate ↑ | Avg_L ↑ | Attack_Rate ↑ |
|---|---|---|---|---|---|---|---|---|---|---|
| StarGAN | 0.211 | 98.8% | 0.222 | 99.2% | 0.259 | 99% | 0.223 | 99.4% | 0.244 | 99.8% |
| AttGAN | 0.108 | 56.9% | 0.114 | 60.7% | 0.108 | 70.5% | 0.112 | 59.5% | 0.123 | 64.1% |
| STGAN | 0.073 | 58.2% | 0.080 | 62.2% | 0.149 | 80.4% | 0.074 | 61.2% | 0.085 | 71.8% |
| U-GAT-IT | 0.242 | 97% | 0.261 | 98% | 0.303 | 100% | 0.267 | 98% | 0.276 | 99% |

The results show that the generated adversarial examples achieve certain effects on each domain of the four deepfake models. StarGAN and U-GAT-IT are relatively vulnerable to adversarial attacks because the *average L* values is much greater than the threshold and the success rate of attack is close to 100%, respectively. The success rates of attacks on AttGAN and StarGAN are relatively lower; AttGAN and StarGAN are relatively less affected by the adversarial attack.

In addition, comparing the three groups of experiments (a), (b) and (c), we see the attack effect can be improved by relaxing the limit of algorithm parameters, such as increasing $\varepsilon$ or $T$ (at the expense of a more obvious perturbation or a larger computational cost). Comparing the three groups of experiments (a), (d) and (e), we find that using a better adversarial attack algorithm (the MI-FGSM is an improvement on the I-FGSM and APGD is an improvement on the MI-FGSM) can improve the attack effect. Both the MI-FGSM and APGD perform $J$ gradient backpropagation in each iteration, which is the same as the I-FGSM. Therefore, they have the same algorithm time complexity O($T$) and roughly similar computational cost. All this shows that the CDMAA framework is well compatible with adversarial attack algorithms, and the general improvement methods are also applicable to CDMAA.

Figure 4 shows the attack effectiveness of some of the adversarial examples in the above experiment:

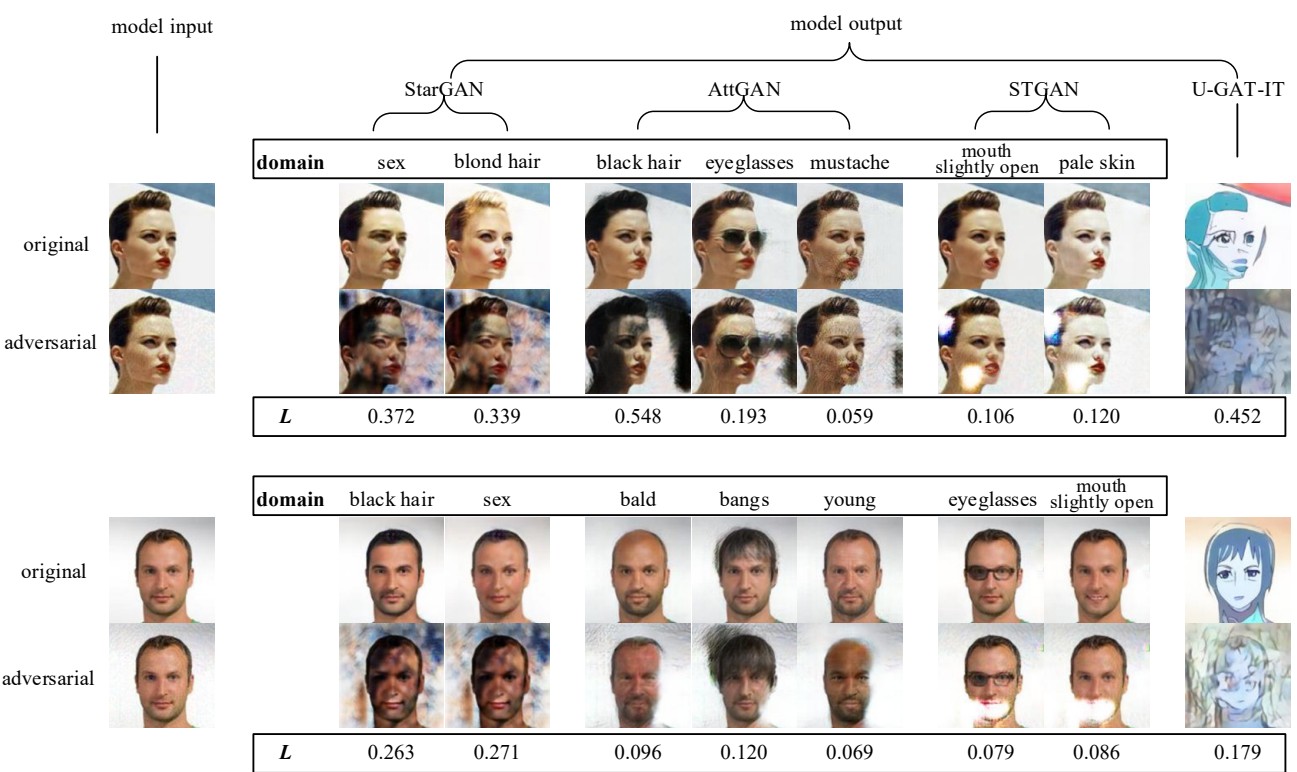

**Figure 4.** Illustrations of adversarial attack. The first column shows the original images and its adversarial examples, and the next eight columns illustrate the output images of models in corresponding domains with the first column as the input.

Figure 4 shows that the difference between the adversarial example and the original image is so small that the human eye can hardly distinguish it. However, the difference between the output of the deepfake model, i.e., the distortion of the fake image, is large enough to be distinguished. Therefore, using the adversarial example instead of the original image can significantly deform the output of the deepfake model so as to effectively prevent the model from forging pictures.

### 4.3. Ablation/Contrast Experiments

#### 4.3.1. Cross-Domain Gradient Ablation/Contrast Experiment

To verify that the method of uniformly weighted cross-domain gradients used by CD-MAA can effectively expand the generalization of adversarial examples between various domains, we carry out the contrast attack experiment, where we keep other components of CDMAA unchanged and only change the way to handle different gradients in domains:

(1) Single gradient: $\boldsymbol{grad}^{(j)} = \boldsymbol{grad}_1^{(j)}$, i.e., use the gradient of only one domain as the cross-domain gradients, without considering the generalization of the generated adversarial examples in other domains. This problem exists in most current studies [30,32,33]. (2) **Iterative gradient:** $\boldsymbol{grad}^{(j)} = \boldsymbol{grad}_{t \bmod K_j}^{(j)}$, i.e., iteratively use the gradient of each domain loss function as the cross-domain gradients [13]. (3) The **MGDA**: $\boldsymbol{grad}^{(j)} = \underset{i=1,\dots,K_j}{MGDA}(\boldsymbol{grad}_i^{(j)})$, i.e., use the MGDA to generate cross-domain gradients. The results are shown in Table 2:

**Table 2.** The effect of different cross-domain gradient calculation methods on adversarial attack. The last group "Uniform weighting" is used in CDMAA and the other three are contrast groups.

| | Single Gradient | | Iterative Gradient | | MGDA | | Uniform Weighting | |
|---|---|---|---|---|---|---|---|---|
| | *Avg_L* ↑ | *Attack_Rate* ↑ | *Avg_L* ↑ | *Attack_Rate* ↑ | *Avg_L* ↑ | *Attack_Rate* ↑ | *Avg_L* ↑ | *Attack_Rate* ↑ |
| StarGAN | 0.253 | 99.4% | 0.207 | 98.8% | 0.493 | 100% | 0.200 | 100% |
| AttGAN | 0.049 | 29.2% | 0.067 | 48.4% | 0.067 | 42.6% | 0.094 | 55.4% |
| STGAN | 0.064 | 47.6% | 0.060 | 48.2% | 0.057 | 53.6% | 0.063 | 48.6% |
| U-GAT-IT | 0.364 | 99% | 0.242 | 97.0% | 0.742 | 100% | 0.220 | 100% |

Figure 5 shows the visual comparison of the result. Compared with existing research on adversarial attacks against deepfake, which only use single domain gradients or iterative gradients in each domain, the CDMAA framework, using the method of uniform weighting to generate cross-domain gradients, can achieve a higher attack success rate against each model, especially those with more domains, such as AttGAN, and effectively increase the generalization of adversarial attack examples between domains. The *average L* of some models using the method of single gradient or iterative gradient can exceed CDMAA, which shows that the effectiveness of the adversarial examples generated by these two methods on each domain varies greatly, which is not as well-balanced and stable as CDMAA. In addition, compared with using the MGDA to generate cross-domain gradients, the effectiveness of simply using uniform weighting is not quite different, but it can greatly reduce the time consumption (Section 3.5). Therefore, CDMAA uses the most efficient uniform weighting method to calculate the cross-domain gradients.

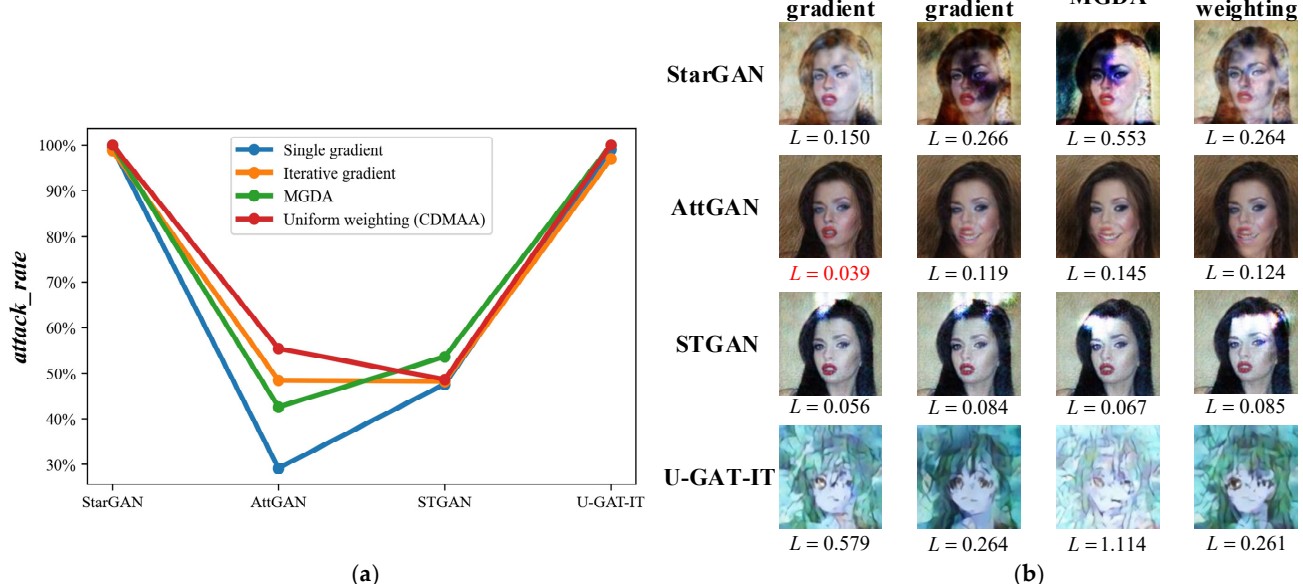

**Figure 5.** Visual comparison of (**a**) *attack_rate* and (**b**) model output disruption for different cross-model gradient calculation methods. The values of *L* below 0.05 in (**b**) have been marked red. Combining the attack rate of four models into consideration, "Uniform weighting" is superior to "Single gradient" and "Iterative gradient" and not far off from "MGDA".

4.3.2. Cross-Model Perturbation Ablation/Contrast Experiment

To verify that CDMAA uses the MGDA to calculate the cross-model perturbation vector $w$, which can effectively expand the generalization of adversarial examples between various models, we carry out the contrast attack experiment, where we keep other components of CDMAA unchanged and only change the way to process each cross-domain gradient: (1) Single gradient: $w = grad^{(1)}$, i.e., only use the cross-domain gradients of one model to update the adversarial example, which is equivalent to ignoring whether the adversarial example has the generalization ability to attack other models. (2) **Iterative gradient**: $w = grad^{(t \bmod J)}$, i.e., iteratively use the cross-domain gradients of each model to update the adversarial example [12]. (3) **Uniform weighting**: $w = \frac{1}{J}\sum_{j=1}^{J} grad^{(j)}$, i.e., use the mean of the cross-domain gradients of each model to update the adversarial example [34]. The results are shown in Table 3:

**Table 3.** The effect of different cross-model perturbation calculation methods on adversarial attack. The last group "MGDA" is used in CDMAA and the other three are contrast groups.

| | Single Gradient | | Iterative Gradient | | Uniform Weighting | | MGDA | |
|---|---|---|---|---|---|---|---|---|
| | *Avg_L* ↑ | *Attack_Rate* ↑ | *Avg_L* ↑ | *Attack_Rate* ↑ | *Avg_L* ↑ | *Attack_Rate* ↑ | *Avg_L* ↑ | *Attack_Rate* ↑ |
| StarGAN | 1.143 | 100% | 0.851 | 100% | 1.176 | 100% | 0.199 | 99.4% |
| AttGAN | 0.000 | 0.0% | 0.033 | 18.3% | 0.000 | 0.0% | 0.099 | 59.7% |
| STGAN | 0.000 | 0.0% | 0.014 | 7.0% | 0.000 | 0.0% | 0.068 | 57.2% |
| U-GAT-IT | 0.027 | 1.0% | 1.002 | 100% | 1.629 | 100% | 0.216 | 99% |

Figure 6 shows the visual comparison of the results. Although the methods of single gradient, iterative gradient and uniform weighting used in the current research can reach a high *average L* value on some models, such as StarGAN, their effectiveness on models that are robust against adversarial attacks (such as AttGAN and STGAN) are very poor. In fact, it is meaningless to reach such a high *average L* value: as long as the threshold $\tau = 0.05$ is exceeded, the output distortion is obvious enough and a successful adversarial attack is achieved. In contrast, the group "MGDA" can achieve a considerable attack success rate for each model. Therefore, we use the MGDA to calculate cross-model perturbation to improve the generalization of adversarial examples across models.

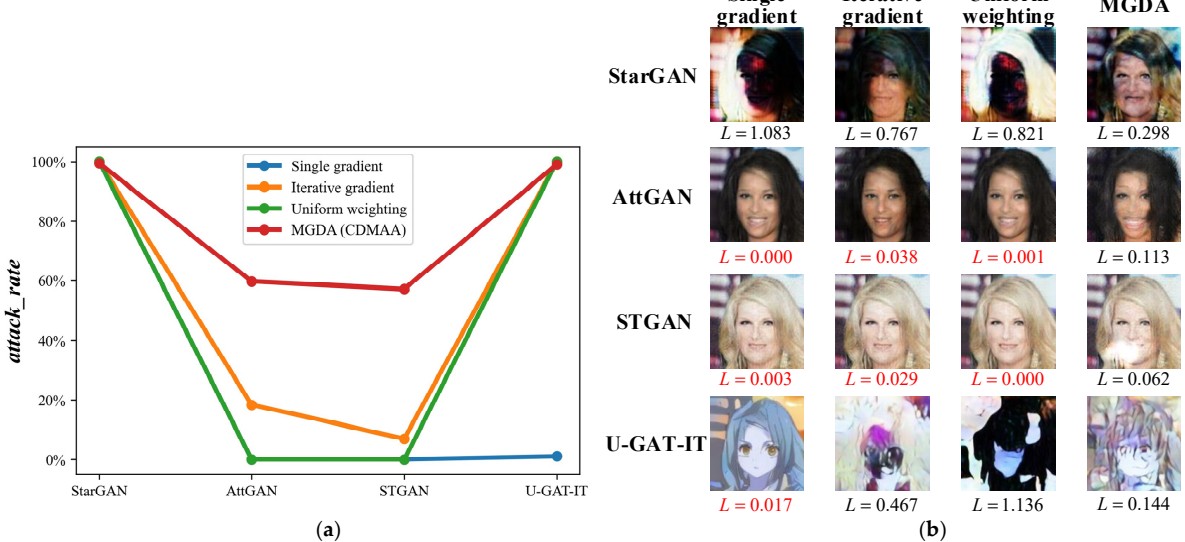

**Figure 6.** Visual comparison of (**a**) *attack_rate* and (**b**) model output disruption for different cross-model perturbation calculation methods. The values of *L* below 0.05 in (**b**) have been marked red. The last group "MGDA" can attack each model well, while the group "Single gradient" has a strong attack effect on only one model and the other two groups have limited attack effects on AttGAN and STGAN.

### 4.3.3. Gradient Regularization Ablation/Contrast Experiment

To verify the effectiveness of gradient regularization used in CDMAA, we carry out the attack contrast attack experiment, where we keep other CDMAA components unchanged and only change the gradient regularization method used: (1) Without regularization: $S^{(j)} = 1$, i.e., the regularization factor is always 1, which is equivalent to not using a regularization method. (2) *L2* regularization: $S^{(j)} = \|\boldsymbol{grad}^{(j)}\|_2$. (3) Logarithmic gradient regularization [15]: $S^{(j)} = L^{(j)}$. The results are shown in Table 4:

**Table 4.** The effect of different gradient regularization on adversarial attack. The last group "Penalty-based gradient regularization" is used in CDMAA and the other three are contrast groups.

| | Without Regularization | | L2 Regularization | | Logarithmic Gradient Regularization | | Penalty-Based Gradient Regularization | |
|---|---|---|---|---|---|---|---|---|
| | *Avg_L* ↑ | *Attack_Rate* ↑ | *Avg_L* ↑ | *Attack_Rate* ↑ | *Avg_L* ↑ | *Attack_Rate* ↑ | *Avg_L* ↑ | *Attack_Rate* ↑ |
| StarGAN | 0.304 | 99.8% | 0.967 | 100% | 0.626 | 100% | 0.221 | 100% |
| AttGAN | 0.119 | 59% | 0.078 | 46% | 0.122 | 60.6% | 0.115 | 62.9% |
| STGAN | 0.054 | 44.4% | 0.076 | 60.8% | 0.012 | 6.3% | 0.073 | 62.6% |
| U-GAT-IT | 0.388 | 98% | 1.389 | 100% | 0.963 | 99% | 0.253 | 100% |

Figure 7 shows the visual comparison of the result. On the metric of *attack_rate*, the method of penalty-based gradient regularization is superior to other gradient regularizations. It achieves a more uniform attack effect distribution on each model by reducing the effect on the model with large loss function value in exchange for a major attack on the model with small loss function value. In the actual process of adversarial attacks, due to the large gap in the vulnerability of each model to attacks, the use of this gradient regularization method will be more practical.

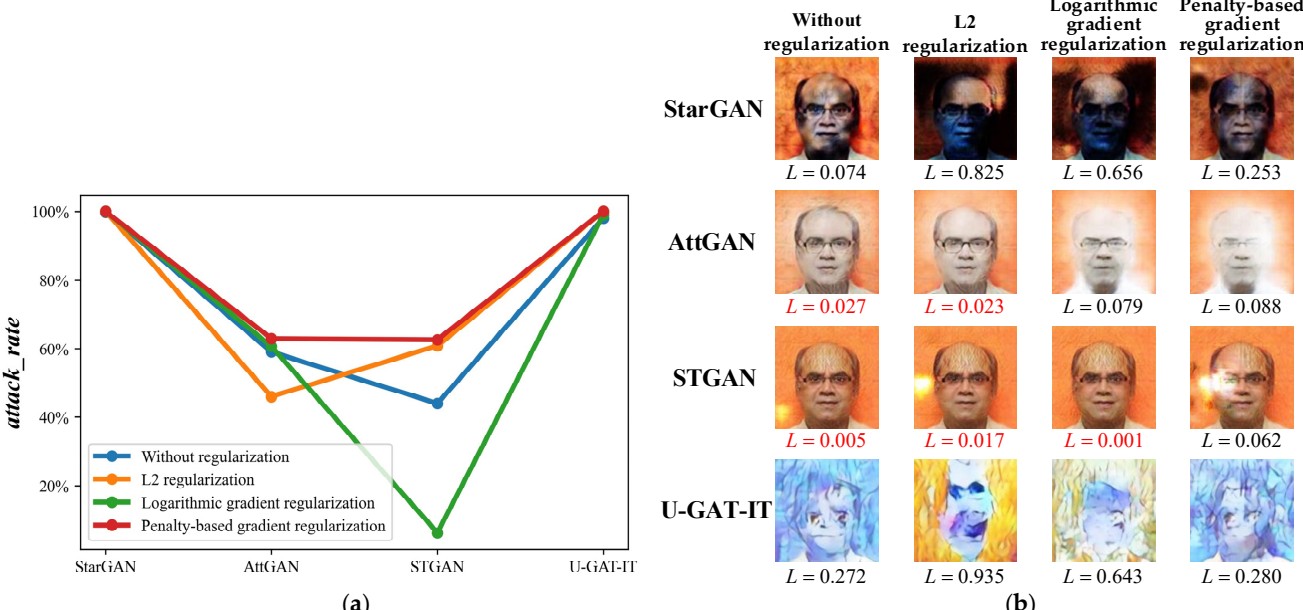

**Figure 7.** Visual comparison of (**a**) *attack_rate* and (**b**) model output disruption for different gradient regularization. The values of *L* below 0.05 in (**b**) have been marked red. The group "Penalty-based gradient regularization" can achieve higher *attack_rate* than the other three groups over four models.

## 5. Conclusions and Future Work

In this paper, we propose a framework of an adversarial attack against the deepfake model called CDMAA, which can expand the generalization of the generated adversarial examples in each domain of multiple models. Specifically, using CDMAA to generate adversarial examples can distort fake images, i.e., the output of multiple deepfake models under any condition variables, so as to interfere with the deepfake model and protect the

pictures from model tampering. An adversarial attack experiment on four mainstream deepfake models shows that the adversarial examples generated by CDMAA have high attack success rates and can effectively attack multiple deepfake models at the same time. Through ablation experiments, on the one hand, we verify the effectiveness of each CDMAA component; on the other hand, compared with other similar research methods, we verify the superiority of CDMAA.

Since CDMAA needs to use the gradient-based adversarial attack algorithm, future work can focus on how to extend this framework to no-gradients-required adversarial attack algorithms, such as AdvGAN [38] or Boundary Attack [39]. In addition, we will try to extend CDMAA to attack other data types of deepfake, such as video and voice.

**Author Contributions:** Conceptualization and writing—original draft preparation, H.Q. and Y.D.; methodology, H.Q. and T.L.; software, H.Q.; writing—original draft, H.Q. All authors have read and agreed to the published version of the manuscript.

**Funding:** This work was supported by the 2021 Fundamental Research Funds for the Central Universities of PPSUC (NO.2021JKF105) and the World-class Discipline University Construction Funds of PPSUC(NO.2021FZB13).

**Institutional Review Board Statement:** Not applicable.

**Informed Consent Statement:** Not applicable.

**Data Availability Statement:** Data available in a publicly accessible repository.

**Acknowledgments:** We thank Zhiqiang Song and Cheng Yang, who spent much time and effort on reviewing, revising and translating this paper.

**Conflicts of Interest:** The authors declare no conflict of interest.

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
