# Peer review of "The Framework of Cross-Domain and Model Adversarial Attack against Deepfake"

_futureinternet, doi:10.3390/fi14020046_

Round 1

Reviewer 1 Report

There are several approaches (models) to create derived images (deepfake) applicable to several domains.  There exist approaches (termed "adversarial attacks" here) that attempt to protect the images that alter images in imperceptible ways such that the deepfake is visibly corrupted and thus obviously identifiable as a deepfake. The authors point out that most existing approaches are effective for specific domains or models.

The paper proposes an approach termed as "Cross-Domain and Model Adversarial Attack" (CDMAA). The approach weighs the loss function for each domain and obtains the cross-domain gradient; which is used to optimize the "success rate" of "adversarial attacks".  It uses four common deepfake models StarGAN, AttGAN, STGAN, and U-GAT-IT, and uses 18 domains (attributes such as black hair, glasses, slightly opened mouth etc). The loss function L values is used as a measure of the success of the "adversarial attacks".

It is not clear why these four models were chosen: StarGAN, AttGAN, STGAN, and U-GAT-IT. Also the selection of the 18 domains should be discussed.

For those who are new to the field the term "adversarial attack" will cause confusion. Generally the "adversary' is the evil entity and an "attack" is an attempt to cause damage. However the implication of the terms is completely opposite in this paper. It is true that the terms have been used in the related literature, for example in some of the papers cited. However it should not be cited that this paper will be read only by the researchers who are already familiar with the usage. I think the authors should  formally define and explain the terms: "model", domain, "adversarial attack" and the Loss function to orient the reader. I would not have used the term "Adversarial Attack" to describe anti-deepfake manipulation of images.

The mathematics behind the approach is reasonably well explained. The results show that StarGAN and U-GAT-IT are more vulnerable to adversarial attacks with success rated close to 100%.

In this paper only still images were used. can the approach be extended to videos with voice, and how?

Author Response

Thanks for your nice comments on our manuscript. According to your suggestions, we have revised our manuscript and attached a point-by-point letter to you. Please see the attachment "Response to Reviewer 1 Comments.docx".

Reviewer 2 Report

This manuscript reports interesting research to protect images, for example those posted on social media and internet from adversarial attacks by Deepfake models that are variants of the popularly known Generative Adversarial Networks (GANs), which can lead to misinformation or tampered information in the form of images. 

The authors report a theoretically sound and interesting solution to the potential problem that can arise from the misuse of Deepfake models as a framework of adversarial attacks on multiple domains and multiple models that operate simultaneously - called Cross Domain and Model Adversarial Attack (CDMAA). A graphical overview of the framework is presented with example images in Figure 3 to show how it works. The Iterative Fast Gradient Sign Method (I-FGSM) algorithm is applied as part of this framework and the resulting algorithm for CDMAA is presented in Algorithm 1.  

I have a few comments to further improve the presentation of the manuscript. 

1. Results from four adversarial Deepfake models  with the CDMAA framework are presented in Table 1 with a comparison of attack rate and other parameters. The table caption is too simple to get a detailed understanding of the results presented there. A more detailed caption with emphasis on the significance of the values of variables presented should be included.  

2. How do the computational cost of the 4 algorithms presented in Table 1 compare? Also discuss, the underlaying details for the differences in the computational costs after adding those details in the revised version. 

3. How critical is the use of Multiple Gradient Descent method in the framework?  Are there are any alternate approaches that can work, for instance in cases where the gradient is likely to vanish or not accessible? Some comments on these aspects will help improve the scope of the paper in a broader context.  

Author Response

Thanks for your nice comments on our manuscript. According to your suggestions, we have revised our manuscript and attached a point-by-point letter to you. Please see the attachment "Response to Reviewer 2 Comments.docx".
